# Preterm Brain Injury, Antenatal Triggers, and Therapeutics: Timing Is Key

**DOI:** 10.3390/cells9081871

**Published:** 2020-08-10

**Authors:** Daan R.M.G. Ophelders, Ruth Gussenhoven, Luise Klein, Reint K. Jellema, Rob J.J. Westerlaken, Matthias C. Hütten, Jeroen Vermeulen, Guido Wassink, Alistair J. Gunn, Tim G.A.M. Wolfs

**Affiliations:** 1Department of Pediatrics, Maastricht University Medical Center, 6202 AZ Maastricht, The Netherlands; d.ophelders@maastrichtuniversity.nl (D.R.M.G.O.); r_gussenhoven@hotmail.com (R.G.); luise.klein@maastrichtuniversity.nl (L.K.); reint.jellema@maastrichtuniversity.nl (R.K.J.); r.westerlaken@maastrichtuniversity.nl (R.J.J.W.); m.hutten@maastrichtuniversity.nl (M.C.H.); 2School for Oncology and Developmental Biology (GROW), Maastricht University, 6229 ER Maastricht, The Netherlands; 3School for Mental Health and Neuroscience (MHeNS), Maastricht University, 6229 ER Maastricht, The Netherlands; 4Department of Pediatric Neurology, Maastricht University Medical Center, 6202 AZ Maastricht, The Netherlands; jeroen.vermeulen@mumc.nl; 5Department of Physiology, Faculty of Medical and Health Sciences, University of Auckland, Private bag 92019, Auckland 1023, New Zealand; g.wassink@auckland.ac.nz (G.W.); aj.gunn@auckland.ac.nz (A.J.G.)

**Keywords:** preterm brain injury, hypoxia-ischemia, chorioamnionitis, timing, therapeutic hypothermia, stem cells, annexin A1, erythropoietin, biomarkers

## Abstract

With a worldwide incidence of 15 million cases, preterm birth is a major contributor to neonatal mortality and morbidity, and concomitant social and economic burden Preterm infants are predisposed to life-long neurological disorders due to the immaturity of the brain. The risks are inversely proportional to maturity at birth. In the majority of extremely preterm infants (<28 weeks’ gestation), perinatal brain injury is associated with exposure to multiple inflammatory perinatal triggers that include antenatal infection (i.e., chorioamnionitis), hypoxia-ischemia, and various postnatal injurious triggers (i.e., oxidative stress, sepsis, mechanical ventilation, hemodynamic instability). These perinatal insults cause a self-perpetuating cascade of peripheral and cerebral inflammation that plays a critical role in the etiology of diffuse white and grey matter injuries that underlies a spectrum of connectivity deficits in survivors from extremely preterm birth. This review focuses on chorioamnionitis and hypoxia-ischemia, which are two important antenatal risk factors for preterm brain injury, and highlights the latest insights on its pathophysiology, potential treatment, and future perspectives to narrow the translational gap between preclinical research and clinical applications.

## 1. Patterns of Preterm Brain Injury

### 1.1. Diffuse White Matter Injury

At present, the most common type of brain injury in preterm infants is diffuse white matter injury (dWMI) [1]. Historically, cystic periventricular WMI or leukomalacia (cPVL), characterized by large cystic lesions in the deep periventricular white matter, was the most common type of white matter injury in preterm babies. Fortunately, this type of white matter injury has become much less common [2], presumptively due to improved obstetric and neonatal healthcare [3]. However, more diffuse types of white matter injuries are increasingly recognized [1,4]. This diffuse WMI is clinically characterized by either diffuse microscopic, punctate lesions, decreased white matter volume, and the thinning of the white matter tracts [5,6].

In humans, cortical myelination begins at approximately 34 weeks of gestation and progresses in a caudal-to-rostral fashion [7,8]. However, cortical myelination is still incomplete at full term, and axonal myelination progressively increases over the first few years of life [9]. Intra-cortical myelin continues to increase until approximately 30 years of age [10,11]. These myelinated axons support brain connectivity by enabling rapid action potential transmission and providing axonal protection.

The predominant cells responsible for myelination are the oligodendrocytes (OLs). OLs undergo critical developmental maturation during the third trimester of pregnancy that renders them particularly vulnerable for perinatal insults [12]. OLs originate from neuronal stem cell (NSC)-derived oligodendrocyte precursor cells (OPCs) that differentiate into immature pre-myelinating OLs (pre-OLs), and finally into mature OLs that, upon contact with axons, produce myelin that encapsulates these axons [12]. In particular, the pre-OLs, that are the most abundant cell type from 24–30 weeks of gestation, are considered the key cellular target in preterm brain injury [13]. Pre-OLs exhibit maturation-dependent characteristics, including overexpression of excitatory amino acid receptors and an immature anti-oxidant system that render them vulnerable to injurious events [14,15]. Under physiological conditions, a consistent pool of OPCs is maintained through the continuous proliferation of OPCs. During development, these OPCs then migrate from the subventricular zone into specific white matter regions, where they differentiate into pre-OLs and subsequent mature myelinating OLs. Due to inflammatory/hypoxic insults, pre-OL decrease in number and local OPCs are triggered to proliferate and replenish the OL pool. In chronic diffuse WMI, the maturation of late pre-OLs is believed to arrest at this pre-myelinating stage, contributing to myelination failure after WMI [16]. The precise mechanisms that lead to a maturational arrest are unclear but likely include excessive accumulation of hyaluronic acid produced by reactive astrocytes, chronic microglial activation, and changes in the cell cycle (e.g., cell cycle exit) of oligodendrocyte progenitors [17].

### 1.2. Grey Matter Injury

Increasing evidence shows that diffuse white matter injury leads to disturbances in grey matter structures, including the cerebral and cerebellar cortex, thalamus, hippocampus, and basal ganglia [18]. As in the white matter, grey matter abnormalities are associated with aberrations in neuronal processes such as impaired neuronal dendritic arborization, rather than cell death alone [19,20]. Although data from multiple animal models have now demonstrated that diffuse WMI is not directly correlated with axonal damage [21,22], it has been shown that oligodendrocytes and neuronal function are inherently intertwined. Oligodendrocytes play an integral role in axonal development and function. Mild diffuse loss of white matter reduces the functional integrity of neuronal axons, thus impairing neuronal growth, development and function after preterm birth [20]. Two specific types of neurons, the GABAergic interneurons, and subplate neurons, are of particular interest in the pathophysiology of preterm brain injury. Subplate neurons comprise a transient neuronal cell population, located just below cortical layer 6, and are essential for the development of thalamocortical connections and accurate formation of distinct cortical layers [19]. The GABAergic interneurons within the subcortical white matter establish connections between adjacent neurons and modulate the neural circuitry. These processes are predominant when vulnerability to preterm WMI is the highest, and indispensable in the development and function of cortical networks [23]. After transient hypoxemia in a fetal ovine model, although no significant cell death was seen in the basal ganglia or subplate [23,24], there was a loss of interneurons and disruption of perineuronal nets in the parasagittal cortex [25]. Furthermore, in a similar study, transient hypoxemia was associated with marked impairment of dendritic arborization and functional dysmaturation [23,24], underlining their potential clinical relevance to the pathophysiology of preterm brain injury.

### 1.3. Cerebellar Injury

There is increased recognition of cerebellar involvement in the adverse neurodevelopmental outcomes after preterm birth [26]. Historically, the cerebellum was considered to be solely responsible for motor function, and altered cerebellar function was linked to ataxia. However, there is accumulating evidence that the cerebellum also plays an important role in the high prevalence of non-motor deficits (i.e., cognition, learning, and behavior) in survivors of prematurity [27].

Clinical data show that a disturbed cerebellar structure in prematurely born children can still be detected at school age/adolescence with diffusion tensor imaging (DTI), and is associated with adverse neurodevelopmental outcome [28]. For example, cerebellar cognitive affective syndrome has been linked to prematurity-related cerebellar injury [27,29].

In particular, during the third trimester of pregnancy, the cerebellum exhibits a rapid increase in growth and development [30]. This accelerated growth is characterized by the proliferation and migration of granule precursor cells from the external granular layer (EGL) to the internal granular layer (IGL) of the cerebellar cortex. This process is essential for the structural and functional integrity of the cerebellum [30]. At this stage of development, the cerebellum is particularly vulnerable to insults that can disrupt normal development (e.g., hypoxia-ischemia (HI), inflammation) and the consequences of preterm delivery, including chronic lung injury [31]. Experimental studies in fetal sheep [32] have shown that perinatal asphyxia increased neuronal death and oxidative stress, and reduced cerebellar strata and astrocytes in the preterm cerebellum. These recent findings suggest that cerebellar injury likely contributes to the morbidities associated with ‘encephalopathy of prematurity’ and highlight the importance of protecting the cerebellum against injurious perinatal insults to prevent long-term neurodevelopmental impairment.

## 2. Pathophysiology of Preterm Brain Injury

### 2.1. Intrauterine Infection and/or Inflammation

Chorioamnionitis (i.e., antenatal infection/inflammation) is defined as inflammation of the fetal membranes (chorion and amnion), and histologically characterized by diffuse infiltration of neutrophils into these membranes. Chorioamnionitis is the most common cause of preterm birth (11–40%), and its incidence increases with decreasing gestation age [33,34]. Chorioamnionitis is present in 3–5% of deliveries at term; however, in deliveries between 21–24 weeks of gestation, chorioamnionitis is confirmed in 94% of placentas [35]. Chorioamnionitis is considered to be a polymicrobial syndrome that includes *Mycoplasmas species* (*Ureaplasma species* in particular), *Gardnerella vaginalis*, and *Fusobacteria species* as the most common isolated bacteria, but viral and fungal species are also reported [36]. The routes by which microorganisms invade the amniotic cavity are ascending from the lower genital tract (most common), hematogenous spread through the placenta, accidental introduction via invasive procedures (i.e., amniocentesis/percutaneous umbilical cord blood sampling), contamination via intrauterine contraceptive devices, and retrograde spread via the fallopian tubes [37]. Since the ascending route is the most common infectious route, it is not surprising that microbial invasion of the amniotic cavity is more frequently found in pregnancies with preterm premature rupture of the membranes (PPROM) and with longer duration of labor. However, intact membranes can also be invaded by microbes and cervical insufficiency, twin gestations, meconium-stained amniotic fluid, presence of genital tract pathogens (e.g., sexually transmitted infections, group B Streptococcus, bacterial vaginosis) and idiopathic vaginal bleeding are associated with increased risk of intra-amniotic infection. Chorioamnionitis is an independent risk factor for adverse fetal organ development, including the brain [38]. Specifically, chorioamnionitis is linked to increased risk of intraventricular hemorrhage [39] and neurologic impairment/injury [40,41], including cerebral palsy (CP) [42].

### 2.2. Fetal Immune Response Syndrome

When the fetus is directly exposed to intrauterine inflammation/infection via direct skin contact or swallowing and breathing movements in utero, the fetal immune system responds with the release of pro-inflammatory cytokines that induce a fetal inflammatory response syndrome (FIRS) [33]. FIRS is characterized by an elevation of fetal plasma interleukin-6 (IL-6), a pro-inflammatory cytokine involved in hematopoiesis, maturation of antibody-producing plasma cells, T-cell activation, and differentiation and secretion of acute-phase proteins by the liver. The elevation of fetal plasma IL-6 concentrations is an independent risk factor for severe neonatal morbidity, including intraventricular hemorrhage (IVH) and PVL [43,44].

A study including 2390 extremely preterm infants (<27 weeks’ gestation) showed that fetal exposure to histological and clinical chorioamnionitis was associated with an increased risk of cognitive impairment at 18–22 months of corrected age, compared to infants not exposed to chorioamnionitis or histological chorioamnionitis alone [45]. These combined data suggest that the release of inflammatory cytokines (i.e., FIRS) in the course of intrauterine infection may play a crucial role in the development of preterm brain injury [44]. Fetal systemic inflammation is a known inducer of cerebral inflammatory responses and is associated with adverse neurodevelopmental outcomes [46].

### 2.3. Hypoxia-Ischemia

Perinatal brain injury after severe hypoxic-ischemia (HI), also known as hypoxic-ischemic encephalopathy (HIE), is associated with high morbidity and mortality rates and occurs in 1–8 per 1000 live births in term infants [47]. Interestingly, the incidence of HIE in preterm infants is higher (4–48 per 1000 preterm newborns), indicating that HI plays an important role in the pathophysiology of preterm brain injury [48,49]. It has been shown that in term-born infants, the damage is directly at the level of the neurons (grey matter), whereas in the premature infants, the damage is occurring more at the level of the white matter with secondary damage to the grey matter [50].

Conceptually, brain injury caused by HI can be described in three phases [51]. The initial phase (primary energy failure) during which there is a decreased supply of oxygen and brain cells switch to anaerobic metabolism, which generates lactate and much less adenosine triphosphate (ATP) per molecule of glucose. As ATP becomes depleted and membrane pumps start to fail, there is accumulation of intracellular sodium and calcium and cytotoxic edema, with the release of excitatory neurotransmitters (especially glutamate) and free fatty acids from degrading cellular structures. In turn, glutamate activates *N*-methyl-d-aspartate (NMDA) and α-amino-3-hydroxy-5-methyl-4-isoxazolepropionic acid (AMPA) receptors on pre-oligodendrocytes and neurons, resulting in reactive oxygen and nitrogen species (ROS and RNS) production. After reperfusion, excitatory amino acid levels resolve and there is apparent recovery of cerebral oxidative metabolism. Although this latent phase is relatively short (typically lasting for approximately 6 h), the evolving neurotoxic cascade is still reversible. Consequently, we consider this critical period as the key “therapeutic window of opportunity” for many neuroprotective therapies [52].

Although the restoration of blood and oxygen supply to the brain limits HI brain damage, reperfusion exacerbates cell injury in the secondary phase. During this phase, that typically extends from 6 to 72 h after the insult, restored reperfusion leads to induction of an inflammatory reaction in the brain and subsequent synthesis of ROS, mitochondrial dysfunction, and persistent glutamate excitotoxicity that eventually results in cell death. Cell death mechanisms include apoptosis, autophagy, and necrosis. Whereas apoptosis and autophagy are controlled paths to cell death through intracellular programming, early-onset necrosis represents premature cell death via unregulated degradation of cell components. In turn, these cell components function as danger-associated molecular patterns (DAMPs) that trigger a sterile (non-infectious) cerebral inflammatory response, characterized by aberrant microglia activation that ultimately can lead to altered brain development and injury. Targeting these processes in this secondary phase of HI-driven preterm brain injury may offer an additional window of pharmaceutical opportunity. In addition, combining interventions in the primary and secondary phases might allow synergistic pharmacological effects. The potential neuroprotective therapies within this context are further outlined below.

Following the secondary phase, there are persistent mechanisms that prevent endogenous repair remodeling and re-organization, which potentially lasts weeks to years. During this tertiary phase of brain injury, epigenetic changes and chronic inflammation sensitize the brain to further injury [53].

## 3. Preterm Brain Injury—Timing Is Key

### 3.1. Antenatal Inflammation

Previous studies of short-term antenatal exposure to different microbial stimuli (inclusing gram negative lipopolysaccharide (LPS), *Ureaplasma Parvum* (UP) and *Candida Albicans*) in preterm ovine lambs indicate that timing rather than the specific inflammatory trigger has a greater impact on abnormal histological outcome in the fetal brain [54,55]. This critical role of timing was recently confirmed and explored in an LPS kinetics study. In this study, we found that intra-amniotic exposure to LPS induced a rapid systemic and neuroinflammatory response within 12 h after LPS exposure, whereas white matter injury was not detected until 15 days post-LPS exposure [56].

The mechanisms potentially responsible for this impaired white matter development, amongst others, include excessive microgliosis, loss of metabolic support, and disturbed glutamate homeostasis [57,58], resulting in subsequent cell death, and then the maturational arrest of newly formed preOLs that impair the developing immature brain. Moreover, inflammation-induced epigenetic changes during early development can lead to substantial lasting neurodevelopmental impairments later in life [31,32]. These findings highlight that the pathophysiological and treatment window after endotoxin-induced preterm brain damage is both protracted and dynamic.

As we discuss later, multiple experimental studies have identified recombinant erythropoietin (EPO) as a potential neurotherapeutic after inflammation and HI-induced perinatal brain damage, with inconsistent outcomes reported in clinical studies. Within this context, we investigated the temporal expression of the EPO receptor (EPO-R) in the white matter, cortex, and hippocampus of LPS-exposed preterm fetal sheep. We found that the total EPO-R expression remained unchanged following intra-amniotic LPS exposure. However, the expression of the phosphorylated EPO receptor (pEPO-R), representing its activated form, displayed a distinct time-dependent switch from over-expression to under-expression. Specifically, after 5 h of LPS exposure, the pEPO-R expression was increased in the white matter and cortex. In contrast, at two days post-LPS exposure, a decrease in pEPOR expression was observed that persisted until eight days post-LPS [56]. Such a suboptimal window for EPO treatment, as identified following antenatal infection and potential other perinatal inflammatory hits, might form an explanation for the existing inconsistencies between different clinical EPO studies.

These findings suggest that the timing and onset of inflammation is a key determinant for the type and trigger of pharmacological interventions, which will be discussed below.

### 3.2. Hypoxia-Ischemia

In preterm fetal sheep, global HI caused a potent neuroinflammatory response, characterized by excessive microglial activation, from 3 h [59] until 21 days post-HI [60,61,62,63,64,65,66,67,68,69,70,71,72,73]. This was associated with acute loss of pre-OLs at 72 h after the insult, and restoration of pre-OL numbers at seven days and 14 days post-reperfusion, in association with the increased restorative proliferation of oligodendrocyte progenitors [63,74,75]. Importantly, myelinating OLs remained significantly reduced at these time points, suggesting maturational arrest contributing to hypomyelination [16,63,75,76]. These results were recently extended by van den Heuij et al., who demonstrated persistent hypomyelination at 21 days after HI, with a significantly increased ratio of immature to mature OLs, despite no change in the total number of (Olig-2 positive) oligodendrocytes [73].

Importantly, although hypothermia is also protective in preterm fetal sheep after HI, early initiation of therapeutic hypothermia as early as possible within the first 6 h appears to be essential for optimal neuroprotective effects, again underscoring the concept that timing is crucial for pathophysiological changes and concomitant treatment [77].

### 3.3. Blood-Brain Barrier

In both acute HI and chronic neuroinflammation, the disruption of the blood-brain barrier (BBB) is increasingly recognized as an important contributor to brain injury [78]. Critically, BBB disruption allows inflammatory mediators, such as cytokines and inflammatory cells to enter the brain parenchyma, thereby contributing to further disease progression.

In a preterm ovine model of cerebral ischemia, Malaeb et al. showed increased expression of claudin-5, decreased expression of ZO-1 and -2, but stable levels of occludin and claudin-1 at 72 h after ischemia [79]. These results were complemented by Chen et al., who demonstrated impaired BBB function at 48 h post-ischemia [80]. However, the peak of BBB permeability occurred at 4 h post-reperfusion, suggesting that BBB function improves relatively soon after injury [80]. In an ovine model of global HI, we recently extended these findings by showing albumin leakage and a temporal depletion of Annexin A1 (ANXA1) expression in the cerebrovasculature at 24 h post-reperfusion [81]. ANXA1 is a downstream effector molecule of glucocorticoids with pro-resolving properties, and ANXA1 in the endothelial cells of the BBB promotes cytoskeletal stability and enhances tight junction formation [81]. In addition, we observed upregulation of endogenous ANXA1 in microglia after the HI indicating an additional function of ANXA1 in the microglial response to injury [81]. This finding fits the concept that ANXA1 induces non-inflammatory phagocytosis of cell debris and promotes an anti-inflammatory microglial phenotype that contributes to tissue repair [82,83,84].

Moreover, triggered by the dynamic cerebrovascular ANXA1 changes following global HI, we studied the time-dependent ANXA1 expression in the aforementioned LPS kinetics cohort. Intra-amniotic LPS exposure resulted in a decreased ANXA1 expression in the cerebrovasculature at two and four days post-LPS exposure (Figure 1, original data). Interestingly, vascular ANXA1 expression in the choroid plexus was already decreased at 12 h post-LPS exposure, which might serve as an early gateway for crosstalk between the systemic circulation and the immune-privileged brain.

### 3.4. Immune System Activation

#### 3.4.1. Innate Immune System

Despite differences in etiology (infectious versus sterile inflammation), cross-talk between the peripheral immune system and the immune-privileged brain parenchyma is central in the pathophysiology of preterm brain injury. The biochemical cascade initiated after HI and antenatal infection results in the release of danger-associated molecular patterns (DAMPs) and cytokines (IL-6 in particular) from the circulation contribute to the activation of cerebral endothelium and disruption of the blood-brain barrier, as elegantly reviewed by Mallard et al. [85], thereby undermining central nervous system (CNS) immune privilege and allowing active recruitment of leukocytes that contribute to the neuroinflammatory response, which is primarily mediated by microglia [63,86,87].

Microglia, the resident innate immune cells of the brain, play an important role in brain homeostasis and immune surveillance. They can recognize a wide range of signals (i.e., pathogen-associated molecular patterns and damage-associated molecular patterns) that indicate a threat to the structural and functional integrity of the CNS through various pattern recognition receptors (PRRs), glutamate receptors, and purinergic receptors, resulting in activation [88,89,90]. Upon activation, they produce pro-inflammatory cytokines, chemokines, reactive oxygen species, and excitotoxic molecules [91,92]. Strikingly, the complete depletion of microglia in models of HIE increases brain injury, indicating that microglia are crucial for tissue repair [93,94,95]. Accordingly, the prevention of excessive microglia activation while simultaneously promoting their phagocytic and neurotrophic function can have protective effects. These findings fit well with the concept of type 1 (M1) and type 2 (M2) microglia dichotomy, depending on microenvironmental cues, microglia can rapidly change their phenotype to pro-inflammatory, cytotoxic (type 1) cells, or anti-inflammatory regenerative (type 2) cells, respectively, and intermediate phenotypes [96,97]. Efforts have been made to characterize the dynamics of the microglia response after HI (6 h–7 days), showing that 6 h after HI, there is a mixed response of M1 and M2 markers that normalizes 24 h to seven days after HI [98]. Others report increased (iNOS), specifically in microglia, and a general increase in pro-inflammatory and anti-inflammatory genes within the brain tissue 24 h after an HI event, that were increased by previous sensitization with LPS [99].

Consistent with an acute inflammatory response of resident microglia, clinical studies demonstrated that cytokine levels reach a peak at 12 h after HI [100]. This systemic accumulation of DAMPs immediately after the injurious event has been associated with a massive activation of the peripheral immune system and with a rapid mobilization of immune effector cells (i.e., neutrophils, monocytes, T-cells) from the spleen [91,101]. These mobilized effector cells can invade the neonatal brain through a disrupted blood-brain barrier and aggravate the existing injury [91,101]. The reports aiming to characterize microglia phenotype after HI mostly disregard the contribution of infiltrating monocytes, recruited from the periphery and differentiating into macrophages, to neuroinflammation as macrophages and microglia share a similar expression profile. Similarly, to microglia, macrophages are plastic in phenotype and can either exacerbate brain injury or promote regeneration. There is evidence that monocytes are recruited into the brain after HI as levels of monocyte chemoattractant protein-1 (MCP-1) increase acutely within the brain and in mouse models in which only mature myeloid cells from the periphery express an enhanced green fluorescent protein, fluorescence is detected 24 h and seven days after HI within the CNS [102,103]. In the same model, antibody-mediated depletion of monocytes was neuroprotective but only in male mice. This suggests monocytes contribute to brain damage in male mice [102]. Contrarily, it has been shown that mice pre-conditioned with LPS generated splenic monocytes that protect the brain in a stroke model [104]. It seems that depending on the context, monocytes are a double-edged sword, and influencing their phenotype would be an interesting therapeutic approach.

In a mouse model that expressedgreen fluorescent protein only in myeloid cells, granulocytes infiltrated the brain at 24 h and seven days after HI as well, though they contributed less compared to monocyte infiltration to the total number of infiltrating CD11b+ cells (a marker of macrophages and microglia) [102]. In line with this finding, neutrophils were detected in the cerebrovasculature hours after HI and the depletion of neutrophils before HI reduced brain swelling [63,105,106]. However, when neutrophils were depleted later than 4–8 h after reperfusion, the protective effect of neutropenia was abolished, suggesting an important timing effect of neutrophils in neonatal HI [106].

#### 3.4.2. Astrocytes

In the healthy neonatal brain, astrocytes, along with pericytes, are an important component of the neurovascular unit (NVU) that regulate blood flow, ion balance, support neurons, and exert anti-oxidant actions [107,108,109]. At 25–40 weeks of gestation, astrocytes, predominating the NVU, cover 60% of the cerebrovasculature with their end-feet [108,110]. As previously described, during neuroinflammation, endothelial cells of the BBB become activated, releasing pro-inflammatory cytokines. Astrocytes bare receptors of the innate immune system, secrete matrix metalloproteinases (MMPs), and cytokines of pro- and anti-inflammatory nature, potentiate excitotoxicity through iNOS, increase their expression of the glial fibrillary acidic protein (GFAP), and proliferate and form glial scars after injury [107,111,112,113]. Their immune cell function and close coupling to the cerebrovasculature strongly suggest that they actively contribute to the neuroinflammatory cascade.

Depending on the timing and phase of neuroinflammation, astrocytes could be detrimental or neuroprotective in adult pathologies [113]. Further studies enforced the dual role of astrocytes and termed them A1 and A2 astrocytes by analogy with the M1 and M2 forms of microglia, the first one being most abundantly present in adult neuropathologies [113,114]. In response to neonatal hypoxia-ischemia and interleukin 1-beta (IL-1β)-induced brain injury, A2 astrocytes highly expressing Cyclooxygenase-2 (COX-2) predominatly leading to oligodendrocyte progenitor maturation arrest [115]. The inhibition of the COX-2/prostaglandin E2 pathway improved white matter outcome [115].

Recently, Disdier et al. exposed fetal sheep in utero to prolonged intravenous LPS and assessed changes in the NVU. They found astrogliosis with reactive astrocyte morphology (astrocytic feet-swelling) in the cerebral cortex and white matter, and a decrease in astrocytic vessel coverage in the white matter together with microglia activation which might impair brain maturation and lead to injury [116]. Similarly, we showed in an ovine model of infection-induced fetal neuroinflammation, in which intra-amniotically administered *Candida Albicans* actively invaded the systemic circulation, that astrocyte numbers and GFAP expression increased two and five days after the infection along with white matter injury and microglial activation [55]. By contrast, chronic intra-amniotic exposure of UP decreased astrocyte numbers and resulted in white matter loss in fetal sheep [54]. This decrease in the number of astrocytes after 42 days of UP exposure is important since these cells possess several essential functions in brain development, including the regulation of the extracellular glutamate homeostasis, which provided structural and metabolic support to surrounding cells (e.g., oligodendrocytes) and modulate neuronal connections [57,117]. The changes in astrocyte function or density also result in altered neurological outcomes. In particular, the altered astrocyte (GFAP) protein expression and disrupted astrocyte maturation have been implicated in the pathogenesis of neurodevelopmental disorders such as autism and CP [118,119].

In the context of neonatal HI induced brain injury, astrogliosis is observed with reactive astrocytes exhibiting swollen end-feet [120,121,122]. Moreover, marked astrogliosis was observed at three and seven days after global HI in ovine fetal models [62,123,124]. Importantly, inflammation-induced opening of astrocytic gap junction (connexin) hemichannels, as recently reviewed by Galinsky et al. [125], is a key regulating event in the evolution of oligodendrocyte and neuronal injury. Likely mechanisms include modulation of intracellular calcium handling, blood-brain barrier integrity, purinergic receptor signaling, and inflammasome pathway activation. These cellular processes are likely to initiate a cycle of excessive ATP release, which propagates activation of purinergic receptors on microglia and astrocytes to trigger inflammation-induced injury of neurons and oligodendroglia [125].

#### 3.4.3. Adaptive Immunity

Pathological inflammation can also be mediated by cells of the adaptive immune system. Lymphocytes have been found to contribute to pathophysiology of neonatal brain injury. In particular, T- and B-cells were observed in post-mortem brains of premature infants with PVL and in brains of rodent models for HI three to seven days after HI [63,65,66,126]. Further, lymphocyte deficient mice showed decreased white matter injury after HI [126], and splenectomy prior to HI in neonatal rats resulted in significant neuroprotection [87]. Perinatal chronic hypoxia in mice also led to the infiltration of autoreactive, myelin-specific CD4^+^ T-cells contributing to white matter injury [127]. The contribution of peripheral immune effector cells to brain injury has been confirmed and extended by our data in a preclinical model of neonatal HIE, showing that neuroinflammation mediated by microglia was associated with marked mobilization of the peripheral immune system and splenic involution [63,65]. In addition, we showed that splenic T cell tolerance, induced by mesenchymal stem cell treatment, resulted in decreased cerebral infiltration of T lymphocytes and correlated with white matter protection [65]. However, Herz et al. showed, in a rodent HI model, that peripheral T-cell depletion exacerbates brain injury, suggesting that T-lymphocytes also have neuroprotective roles in the context of neonatal brain injury [128].

## 4. Treatment

Following birth, all preterm infants receive supportive care, including adequate respiratory support, blood pressure support, normalization of fluid-electrolyte balance, and blood glucose levels, and, if necessary, antibiotic treatment, and treatment of seizures. Preterm brain injury evolves over time in a period of hours to weeks, creating a window of opportunity for treatment. However, therapeutic options to improve the outcomes of preterm neonates suffering from brain injury are very limited.

Given the dominant role of inflammation in the induction of preterm brain injury, we consider that immune-modulatory interventions have a high potential to support recovery from brain injury in vulnerable neonates. Immune modulation can be directly effective or indirectly by improving the efficacy of other therapies. A summary of studies on neuroprotective therapies for preterm and near-term neonates is provided in Table 1.

### 4.1. Therapeutic Hypothermia for Preterm Hypoxia-Ischemia

Therapeutic hypothermia is now standard care for near-term to term infants with moderate to severe HIE. Clinical guidelines recommend therapeutic hypothermia should be started as early as possible within 6 h of birth and continued for a period of 72 h, with a target brain temperature of 33.5 ± 0.5 °C [77]. Therapeutic hypothermia is associated with immunosuppressive changes [155], including inhibition of microglial activation, chemotaxis, production of pro-inflammatory cytokines, and nuclear translocation nuclear factor kappa-B [156,157,158]. Confirming extensive preclinical evidence [159], a meta-analysis of 11 randomized controlled trials of systemic and head cooling in infants with HIE (≥ 35-week gestation) showed that hypothermia reduces brain damage on imaging after rewarming [160], and improves survival without disability at 18 to 24 months of age [129]. Long-term follow-up data support improved outcomes, including reduced risk of death, or CP, or an IQ score < 55 in 6–7-year-old children [130,161]. Moreover, hypothermia has had an excellent safety record, with benign physiological effects such as sinus bradycardia, mild thrombocytopenia, and scalp edema during cerebral cooling [129]. Thus, therapeutic hypothermia is a remarkable example of a successful bench to cotside translation.

There are limited clinical evidence for therapeutic hypothermia in preterm (≤ 35 weeks’ gestation) infants. Thus, therapeutic hypothermia is not the standard of care for neonates born very preterm. Preclinical evidence, however, strongly supports a similar pharmacodynamic profile in preterm animals as at term equivalent. In preterm-equivalent fetal sheep, cerebral cooling for 72 h (with extradural temperature titrated to 29.5 ± 2.6 °C) started 90 min after severe asphyxia was associated with basal ganglia and hippocampal neuroprotection, protection of immature OLs in periventricular and parasagittal white matter, and reduced overall microgliosis and apoptosis [74,133]. In association with this histological improvement, higher EEG frequencies recovered faster, cephalic blood flow was restored, and the stereographic seizure amplitude reduced [74]. Importantly, anatomical brain structures that are frequently injured in term and preterm infants with acute HIE were protected.

Historically, systemic-mild hypothermia was associated with reduced survival in extremely low birth weight (≤1000 g) neonates [162]. The well-known systemic effects of hypothermia have raised concerns that induced cooling might promote postnatal hypotension, intracranial bleeding, or respiratory compromise in the extremely preterm infants [163]. In a cohort of neonates cooled outside the standard hypothermia criteria (*n* = 36, e.g., infants at 34–35 weeks’ gestation, or with postnatal collapse, or cardiac disease), compared with infants cooled to the protocol (*n* = 129), complication rates and neurologic outcomes at 18–20 months were similar, except that five newborns who developed intracranial hemorrhage had very poor outcomes [164]. In a retrospective cohort of 31 preterm infants born at 34–35 weeks’ gestation, hypothermia appeared to be associated with increased mortality compared to 32 term infants (12.9% vs. 0%, *p* = 0.04) and a greater prevalence of white matter damage on modern imaging (66.7% vs. 25.0%, *p* = 0.001) [132]. By contrast, a retrospective cohort study of preterm neonates (gestational age range 33–35 weeks) with HIE found death or moderate-severe NDI in 11/22 (50%) cases, similar to rates of adverse outcomes in treated term infants in the large controlled hypothermia trials [129]. Thus, hypothermia is clearly feasible in preterm infants, and appropriately powered controlled clinical trials are essential. One multi-center randomized controlled trial is currently in progress testing safety, feasibility, and efficacy of therapeutic hypothermia started within 6 h of birth in preterm infants at 33–35 weeks’ gestation with moderate to severe HIE (ClinicalTrials: NCT01793129).

One factor that might affect response to therapeutic hypothermia is that preterm newborns have high rates of prenatal and postnatal infection/inflammation [165]. There is some evidence that suggests a possible deleterious interaction between infection and induced hypothermia. For example, a randomized clinical trial in adult patients diagnosed with bacterial meningitis reported that cooling (32–34 °C, for 48 h) was associated with excess mortality (51% vs. 31%), compared to normothermic patients [166]. In neonatal p7 rats subjected to HI, hemispheric and hippocampal protection with systemic cooling was lost after pre-insult sensitization with gram-negative LPS [134] but not gram-positive PAM3CSK4 [135], suggesting a pathogen-dependent effect. The potential effect of infection in encephalopathic infants treated with hypothermia is unclear, as known infection was an exclusion criterion in the hypothermia trials [129].

### 4.2. Cell-Based Therapies

#### 4.2.1. Stem Cells

Cell-based interventions as a therapeutic strategy for injury to the (neonatal) brain have attracted much attention in the past decade [136,167,168,169,170,171,172,173,174]. Many different types of stem cells, derived from fetal, placental, and adult tissues, are currently under investigation [175,176,177,178,179,180]. Stromal cells, including mesenchymal stem cells and multipotent adult progenitor cells (MAPC) are a subset of progenitors that have been shown to differentiate into multiple lineages (i.e., osteoblasts, adipocytes, and chondrocytes) [175,176,177,178]. They have been a particular focus of research as they are easily obtainable (e.g., from cord blood, Wharton’s jelly and bone marrow) and do not have the ethical and safety concerns of embryonic stem cells [172,174]. Moreover, bone marrow-derived adherent stromal cells have low immunogenicity due to a lack of expression of MHC class II antigens, allowing their use for allogenic therapy [174]. The therapeutic potential of mesenchymal stem cells (MSCs) has mainly been attributed to their immune-modulatory and regenerative potential [172,173,177,181]. MSCs modulate innate and adaptive immune responses from a pro-inflammatory status towards an anti-inflammatory status, thereby reducing tissue injury and creating an environment supporting tissue repair and regeneration [177,178,182].

Besides immune modulation in favor of regeneration, MSCs directly affect the injured CNS through secretion of neurotrophic factors that stimulate and maintain neurogenesis of the endogenous neural stem cell population and its subsequent differentiation into neuronal and oligodendroglial lineages [173,177,181,183,184,185].

The neuroprotective potential of systemically administered MSCs and Multipotent Adult Progenitor Cells (MAPC) was shown in a preclinical preterm ovine model of global hypoxia-HI injury [65,67]. MSC administration during the latent phase following global HI resulted in functional improvement and prevented hypomyelination of the subcortical white matter at seven days post HI. These protective effects were attributed to anti-inflammatory effects since neuroinflammation, and peripheral immune activation were significantly reduced [65,67]. Specifically, intravenous MSC treatment reduced the proliferative capacity of splenic T lymphocytes with concomitant reduced cerebral T cell infiltration. In addition, systemic administration of MSCs ameliorated splenic involution caused by global HI, implicating a key role of the spleen in the protective mechanisms of stem cell therapy. The results from this study were supported and extended by studies from Walker et al. [186,187] and Depaul et al. [188], demonstrating a crucial role for the spleen in (1) the pathophysiology of CNS injury and (2) providing mechanistic insight for the beneficial effects of stem cell therapy. Moreover, there is increasing evidence for the effectiveness of stem cell administration via the intranasal route, which is a feasible route for the pharmacological treatment of neonates. This concept of intranasal administration is based on the passage of stem cells over the cribiform plate along the olfactory nerve, allowing for rapid dispersion throughout the brain, where local effects can be exerted [73,189,190,191,192,193]. In a study by van Velthoven et al. delayed (10 days post HI injury, induced at P9), nasal administration of MSC therapy improved neurological outcomes 28 days post HI. Interestingly, improved outcomes were attributed to the regenerative potential of MSC rather than inhibition of injurious processes or prevention of injury since cells were administered when injury was readily established [192]. The delayed intravenous MSC administration in fetal sheep five days after global HI was less effective than acute administration 12 h after reperfusion [72]. These results were attributed to systemic and neuro-immunomodulatory effects [72]. Collectively, the administration route in relation to the timing of administration appears to be crucial for the action of MSCs (immune modulation and regeneration) to establish optimal neuroprotection. As such, repeated dosing by different administration routes throughout different phases (latent, secondary, and even tertiary) of disease progression might ultimately lead to the greatest benefit.

#### 4.2.2. Clinical Stem Cell Trials

Therapeutic hypothermia has improved intact survival and neurodevelopmental outcome in infants with moderate-severe neonatal encephalopathy [129,130,131]. However, neuroprotection is partial, and a significant proportion of asphyxiated infants still die or suffer life-long consequences, including CP, cognitive deficits, and epilepsy [129,130,131]. Encouragingly, preclinical and clinical studies are now investigating therapeutic hypothermia as a potential treatment for extending over several days after the insult infants with mild encephalopathy [194]; however, hypothermia is counter-indicated for preterm infants as discussed above, and there is currently no neurotherapeutic treatment for these patients. Critically, additional therapeutic strategies that can augment the neuroprotective effect of therapeutic hypothermia in infants with moderate-severe encephalopathy, and reduce the neurological burden in (extreme) preterm infants are under active investigation [167]. In light of our discussion on chorioamnionitis, the potential adjuvant therapies should target the detrimental neuroinflammatory response that occur during and after the reperfusion phase after birth asphyxia, while also promoting neuronal regeneration. As reviewed above, MSCs and MAPC meet those criteria and are emerging as a promising therapeutic intervention for multiple diseases in neonatal medicine. A recent clinical trial in adult ischemic stroke patients showed that MSCs [195] and MAPC cells were safe and improved neurological symptoms after one year of follow-up [195,196].

More recent clinical trials have also demonstrated that stem cells are safe, with promising protective results in multiple neonatal diseases, including HIE [197], bronchopulmonary dysplasia (BPD) [198,199,200], IVH [137,138], and established CP [201].

Within this context, recent findings indicated that MSCs may interact either synergistically [202] or antagonistically [203] with therapeutic hypothermia in experimental models of neonatal HIE, depending on the timing of administration. These conflicting findings indicate that the translational gap between preclinical research and the clinical application still needs to be addressed before stem cells could be safely translated into clinical practice for neonatal diseases. Several clinical trials testing the protective potential of stem cells are still recruiting or pending, including for NCT04255147 for BPD, NCT03356821 for perinatal stroke, and NCT02612155 for HIE.

#### 4.2.3. Stem Cell Therapy—Timing Is Key

Although a large body of experimental animal studies have demonstrated the beneficial effects of cell-based therapies for preterm brain damage, (pre)clinical studies confirming these data are limited. In part, this mismatch can be attributed to different methodological approaches between animal models and clinical practice in terms of the use of single-hit animal models whilst clinical etiology is multifactorial, use of inadequately characterized and heterogeneous stem cell populations, route of administration, and dosing strategies. At the same time, clinical trials often use top down approaches in which stem cells are administered at trivial time-points and the clinical outcomes are measured. Therefore, future studies testing stem cells should closely align with the underlying pathophysiology and stage of injury to address the multi-factorial nature of preterm brain injury.

In addition, the stem cell secretome, which is modulated by micro-environmental cues during the different phases leading to preterm brain injury, defines efficacy of administered stem cells. Thus, a detailed analysis of the biodistribution of stem cells over time combined with detailed secretome analysis might (1) unravel novel potential pathways involved in the pathophysiology preterm brain injury, (2) and enable adjunctive cell-free therapies comprising specific trophic and immunomodulatory factors, and other regulators (such as miRNA), and most importantly (3) provide insight into timing, which appears to be a crucial determinant for optimal therapeutic efficacy.

#### 4.2.4. Stem Cell-Derived Extracellular Vesicles

Despite that immunomodulatory and regenerative effects have been shown, the underlying mechanisms of action of stem cell therapies remain largely unknown. It was initially thought that the therapeutic action of stem cells relied on direct replacement of dead and injured cells. However, since the number of cells that reach the site of injury is minimal, with marginal engraftment and short cell survival, this theory has been largely discredited [139,188]. Consistent with this, we and others have been unable to identify stem cells within the cerebral parenchyma seven days after intravenous administration [65,67,139]. Meanwhile, there is growing evidence that the beneficial effects of stem cell therapy are mediated at least in part via paracrine mechanisms as studies have shown comparable therapeutic effects in MSC-conditioned medium compared to its cellular component [204,205]. In addition, we and others have demonstrated potent neuroprotective effects with mesenchymal stem cell-derived extracellular vesicles (MSC-EV) in animal models of preterm brain injury [66,140,141,206].

Remarkably, these therapeutic effects could not be explained by the known anti-inflammatory effects of the MSC-EVs, as observed after MSC treatment [66]. This prompted us to focus on alternative explanations for the pharmacologic effects of MSC-EVs, in particular, restoration of the injured BBB after global HI. There is accumulating evidence that the BBB becomes functional during the second trimester [85,207]. Nevertheless, a global HI insult would result in the release of reactive oxygen species and excitotoxic molecules into the extracellular environment (i.e., the direct effects of HI), with increased cytokine release by the peripheral and local innate immune system (secondary inflammatory component), which leads to BBB dysfunction [208]. In turn, increased BBB permeability allows intracranial infiltration of peripheral immune cells (e.g., macrophages, leukocytes, and T-cells) that aggravate white matter injury via the release of pro-inflammatory mediators. Thus, strengthening or restoring BBB integrity by enforcing endothelial cells which would attenuate the degree of white matter injury.

### 4.3. Pharmacological Interventions

#### 4.3.1. AnnexinA1

ANXA1 (37 kDa), formerly known as macrocortin, renocortin, lipomodulin, or lipocortin-1, was first described in the 1980s, and initially known for its anti-inflammatory effects as a downstream mediator of glucocorticoids [209]. It is a calcium-dependent phospholipid-binding protein that has received more research attention in recent years due to its multimodal function that extends beyond suppressing inflammation [210].

Increased BBB permeability in ANXA1 knock-out (KO) mice and progressive loss of endogenous ANXA1 in the cerebrovasculature and plasma of patients with multiple sclerosis led to better understanding of the function of ANXA1 [211]. ANXA1 strengthens BBB integrity by 1) binding of extracellular ANXA1 to the FPR2 (formyl peptide receptor 2), inhibiting Rho A kinase, and thereby stabilizing tight junctions; 2) direct interaction of intracellular ANXA1 with actin molecules, and thereby promoting cytoskeleton stability and tight junction formation between endothelial cells [211,212]. The intravenous administration of MSC-derived extracellular vesicles containing ANXA1 prevented the previously described HI-induced depletion of endogenous ANXA1 in the cerebrovasculature, thereby preventing BBB leakage [81]. Given the time-dependent drop of ANXA1 in the course of antenatal infection, the ANXA1 administration in this infectious context might be a promising therapeutic strategy. Collectively, our data support the notion that the ANXA1/FPR axis is a therapeutic target to treat fetuses exposed to HI and indicate a pharmacological window of opportunity for ANXA1 supplementation.

#### 4.3.2. Cytokine Treatment

Clinical studies and preclinical animal models have reported acute increases in systemic levels of cytokines such as tumor necrosis factor (TNF)-α, IL-1β, and IL-6 after HI [142,143,144,213,214] and intra-amniotic exposure to inflammation [54,56]. High CSF/serum ratios indicate that local production of cytokines within the CNS also contributes to increased cytokine levels [213]. Thus, decreasing the levels of pro-inflammatory cytokines represents a promising strategy to suppress neuroinflammation and ensure neuroprotection. In preclinical fetal ovine models, neutralizing antibodies against IL-1β and IL-6 or pharmacological antagonism of TNF-α have already been tested and shown promising short-term effects [142,143,144,145]. One day after HI, the systemic infusion of anti–IL-6 mAb attenuated BBB dysfunction and decreased cerebral IL-6 levels. Similarly, IL-1β neutralizing antibodies improved BBB integrity, lowered IL-1β levels in the brain, and reduced IL-1β transport across the BBB [144,146]. Further, histological studies have shown that infusion of anti-IL1β decreased short-term ischemic reperfusion-related parenchymal brain injury [145].

The abovementioned studies have shown promising results, but mainly focused on the short-term effects after HI. Determining the long-term effects is crucial to reduce the detrimental processes that occur during the tertiary phase of the brain injury after HI. As IL-1β and IL-6 expression changes throughout gestation, and they fulfill physiologic functions that are not detrimental by definition, caution has to be paid to the right dosing and timing [214].

#### 4.3.3. Recombinant Human Erythropoietin for Preterm Neonates

Recombinant human erythropoietin (rEpo) is a common treatment for anemia in preterm infants and pediatric patients with chronic renal disease. In addition to stimulating erythropoiesis, rEpo has shown anti-apoptotic, anti-inflammation, and anti-excitotoxic effects after HI and infection-induced perinatal brain injury [215,216], suggesting rEpo could be a promising neuroprotectant for encephalopathic infants. In the long-term, it could promote OL and neuronal maturation and replacement [147,148], and so might promote neurorestoration after injury. Compelling preclinical evidence has shown that early administration of rEpo is neuroprotective over a broad dose range, from 1000 to 30,000 IU/kg, and that continued exposure to high-dose rEpo is more effective, but that optimal treatment regimens are likely to be paradigm-specific.

For example, after maternal LPS injection in rats at 18–19 days gestation, peripheral rEpo (5000 UI/kg) injection at birth was associated with reduced IL6, IL1, and TNF-α concentrations, and apoptosis and demyelination at p7 [217]. Critically, after HI in P7 rats, repeated rEpo injections (5000 IU/kg, at days 1, 2, and 3) provided greater protection than either a single (5000 IU/kg) dose or three injections with 2500 or 30,000 IU/kg rEpo [149], but brain protection was largely lost when the treatment was delayed until 1–3 h after HI [150]. By contrast, P5 mice treated with ibotenic acid (an excitotoxin) had fewer white matter lesions after a single rEpo injection (5000 IU/kg, at 1 h), but additional injections did not augment protection [151]. For post-HI treatment, these data are consistent with the experience with therapeutic hypothermia where optimal protection was achieved when brain cooling was started as soon as possible during the latent phase and continued until the delayed secondary events, such as overt seizures had resolved, after ~72 h, with loss of protection if the treatment is delayed more than ~6 h after global ischemia. Supporting this concept, in preterm-equivalent fetal sheep, prolonged rEpo (5000 IU/kg, from 30 min to 72 h) infusion after asphyxia was associated with partial subcortical white matter and neuroprotection, and improved electrophysiological EEG recovery, in association with reduced apoptosis and inflammation [62]. Similarly, intravenous rEpo (5000 IU/kg) boluses administered once daily for three days to preterm fetal sheep with endotoxin-induced brain damage, reduced axonal damage, microglial, and astrocytic responses in white matter, and improved myelination [152].

There is strong clinical evidence that rEpo treatment is safe in neonatal cohorts and shows indicative evidence of possible benefit. A meta-analyses in preterm infants suggested that rEpo/Darbepoetin started within eight days from birth reduced rates of IVH, PVL, and necrotizing enterocolitis, and improved neurologic deficits, without adverse effects, at 18–22 months’ corrected age (four randomized controlled trials, 1130 cases, relative risk (RR) 0.62, 95% Confidence interval (CI) 0.48 to 0.80; risk difference −0.08, 95% CI −0.12 to −0.04) [218]. However, disappointingly, a more recent large multi-center, randomized, trial that assessed high-dose rEpo treatment for perinatal brain damage in extremely preterm infants (741 patients, 24 to 27 weeks and six days) found that repeated rEpo (1000 IU/kg, i.v.) doses at 48 h intervals for a total of six doses, followed with maintenance doses (400 IU/kg, s.c.) three times per week through 32 completed weeks of postmenstrual age, did not reduce death or severe neurodevelopmental impairment, compared to placebo (26% vs. 26%, RR 1.03, 95% CI 0.81–1.32, *p* = 0.80) at 22–26 months of age [154]. There were also no significant differences between the treatment groups in the rates of intracranial hemorrhage, sepsis, NEC, death, or frequency of serious complications.

These negative results are disappointing; however, there were several limitations that should be kept in mind. First, the optimal rEpo regimen for preterm brain damage is still unknown. Treatment was started from 24 h after birth, which might be too late for benefit. Second, the treatment duration covered the period when PVL is prominent (i.e., 24–32 weeks’ gestation), but it is possible that rEpo treatment should have been continued for longer. There is extensive evidence that chronic microgliosis and extracellular matrix disturbance continue to contribute to disrupted myelination later in gestation, as discussed above [219]. Third, cognitive testing for mild impairments is more sensitive later in life [220], and so potentially long-term follow-up might still show small but beneficial effects. Thus, further studies are required to conclusively determine whether timely and well-targeted treatment with rEpo could alleviate perinatal brain damage in preterm infants.

## 5. Future Perspectives—Closing the Translational Gap

Despite many years of research into preterm brain injury and treatment, there is still an unmet clinical need for patients with preterm brain injury. The, often complex, multi-factorial nature of preterm brain damage, comprising both pre-and postnatal hits, which are not accurately reproduced in current translational models (most models use single hit paradigms), is considered to be paramount for this translational gap.

### 5.1. Multi-Factorial Nature of Preterm Brain Injury

Over the past decade, (pre)clinical evidence demonstrated that the preterm brain could sensitize/desensitize to a second injurious hit after pre-exposure to inflammation [221,222,223]. This concept of preconditioning is supported by clinical data, which shows that the combination of antenatal infection and a hypoxic-ischemic insult around birth dramatically increases the risk of CP (OR 78) when compared to either HI (OR 2.5) or infection (OR 7.2) alone [221]. Besides this specifically combined insult, multiple ante-, peri- or postnatal factors can contribute to the risk of developing brain injury in the preterm infant, including being small for gestational age and having impaired placental growth [224,225,226]. Furthermore, the evidence is accumulating that postnatal ventilation-induced white matter injury, barotrauma around birth, (par)enteral feeding, standard-care medication (glucocorticoids), operative procedures, all may contribute to the development of preterm brain injury [227]. Clinically, exposure to multiple hits is associated with an enormous increase in the risk of and severity of white matter abnormalities [41,224,228]. The combined impact of these factors on preterm brain development is determined by the specific interaction of individual insults and can be modulated by post-insult treatments.

### 5.2. Biomarkers

An important obstacle for the translation of preclinical findings from standardized models is the much higher heterogeneity of the clinical population. Consequently, with the current diagnostic approaches, only a proportion of at risk cases are identified. As such, increasing sensitivity of existing biomarkers, extending the ‘diagnostic toolbox’ with novel biomarkers, or the composition of biomarker panels are essential to identify and stratify preterm infants with brain injury. Improved sensitivity will enable the identification of specific windows of therapeutic opportunity and, therefore, optimize clinical decision-making, leading to more targeted and individualized therapy. Clinical and preclinical evidence for the use of biomarkers to identify preterm brain injury is provided in Table 2.

#### 5.2.1. Plasma Biomarkers

Patient stratification based on reliable biomarkers is key to provide the most optimal treatment strategy. Multiple clinical studies have shown that fetal involvement with the induction of FIRS, characterized by elevated plasma IL-6 levels, during chorioamnionitis correlates with higher rates of neonatal sepsis, multi-organ dysfunction, perinatal brain injury, and CP [43,44,229]. Moreover, increased plasma levels of IL-6 in the acute phase (<24 h) after an ischemic insult or traumatic brain injury have been shown to be a reliable prognostic marker for adverse cerebral outcomes [236,237].

Thus, it is appealing to consider systemic IL-6 levels as a prognostic marker to predict neurodevelopmental outcomes with infection. However, caution is required for several reasons: Firstly, increased plasma IL-6 levels are also observed after non-infection-related insults, including Rh-alloimmunization [238], trauma [239], and HI insults [236]. Secondly, we have demonstrated in the preclinical ovine model that IL-6 concentrations were only transiently increased within the first 24 h after intra-uterine exposure to infection [54,56]. Collectively, the timing of IL-6 detection appears to be essential to interpret IL-6 levels as a marker for cerebral prognosis and so needs to be incorporated into the clinical decision model. Further studies should focus on the use of IL-6 in combination with other promising systemic inflammatory biomarkers (C-reactive protein, procalcitonin).

However, the successful development and implementation of neurotherapeutics in clinical care are currently impeded by a lack of diagnostic tools for early (prenatal) detection and surveillance of intra-amniotic infection. Recently, we demonstrated in a proof of concept that intra-amniotic UP infection induces a distinct time-dependent volatile organic compound (VOC)-signature in the expiration breath of pregnant sheep. Changes in the VOC profile were detectable with good accuracy as soon as 72 h post-infection. Moreover, the VOC profile changed as the duration of the infection progressed [230]. However, these promising results need to be validated in a less standardized clinical cohort, in which the origin of infection is, most often, poly-microbial and confounding factors, such as diet and unbounded environment might influence the sensitivity and specificity of the VOC profile.

#### 5.2.2. Imaging

In contrast to the previously common cystic white matter lesions that were easily detectable by ultrasound and magnetic resonance imaging (MRI), the new diffuse WMI with microscopic, punctate lesions, and altered white matter development are difficult to detect by ultrasound and conventional MRI (T1, T2) [240]. As such, diffuse WMI in preterm infants is detected only in 20% of all cases by conventional MRI at term equivalent age [241], thereby underlining the clinical need for modalities that could accurately and specifically assess brain structure and function in preterm neonates. In recent years, brain-imaging techniques, such as Diffusion Tensor Imaging (DTI) and functional magnetic resonance imaging (fMRI), have allowed scientists to study the brain in more structural detail and detect deteriorating neural networks in the brain [242]. To optimize the sensitivity and specificity of these new imaging modalities, its correlation with pathological changes in the fetal brain is essential. Early therapeutic strategies have been proven to be the most effective in promoting neurodevelopment, thereby highlighting the need for the early prediction of outcome, evaluated by specific measures; MRI [231], especially DWI and DTI, or techniques that could readily be deployed in a neonatal intensive care unit, such as ultrasound [243,244,245], ultrafast Doppler [246], EEG, or functional ultrasound imaging [247] could help develop and translate new treatment strategies and prognostic markers to improve long-term outcomes.

The detection of additional clinical improvement in functional neurological outcomes in the context of therapeutic hypothermia after birth asphyxia requires large numbers of participants. Using predictive neuroimaging and neuromonitoring biomarkers as surrogate endpoints in experimental and proof-of-concept clinical trials may help reduce sample sizes. For example, thalamic n-acetylaspartate (NAA), in term infants, is a highly predictive neuronal marker that could be measured with magnetic resonance spectroscopy (MRS). In asphyxiated and cooled newborns, thalamic NAA acquired days after birth, accurately predicts adverse neurodevelopmental outcome at two years [248,249,250]. Due to its high predictive value and clinical relevance, thalamic NAA may, therefore, be a robust surrogate marker for neurological outcomes in future clinical trials testing the add-on effect of regenerative therapy in addition to therapeutic hypothermia. Since MR imaging is part of standard clinical care in neonates, adding more precise imaging protocols seems obvious and would be feasible. This would also add to the current knowledge mostly based on selected cohorts and may improve the predictive value for early brain MRI.

#### 5.2.3. EEG

Continuous electrophysiological monitoring with conventional (video) electroencephalography (EEG) and amplitude-integrated electroencephalography (aEEG) is an important diagnostic tool in neonatal intensive care [251], especially in the encephalopathic term neonate [252]. However, compared with near-term and full-term newborns, accurate prediction of neurodevelopmental outcomes in preterm neonates with EEG is complicated as electrophysiological characteristics are still developing. Several EEG features have shown promise. For example, Roland sharp waves have been recognized as an early predictor for white matter injury and neurologic outcome [250,251,252]. Further, recent developments in neural network analytics indicate the development of specific cortical networks, and their disruption has been suggested as an early marker of perinatal brain damage [232]. In the high-risk preterm neonate, the suppression of amplitude and high frequencies [232], prolonged inter-burst interval [233], seizures, and disrupted sleep-wake cycling [234,253], are also associated with poor outcomes [153]. EEG and aEEG are generally applied as a diagnostic tool in the neonatal population, even in very premature infants. Currently, the interpretation is based on visual inspection and quantitative measures that are not systematically applied. Since all data are acquired digitally, a systematical quantitative approach can be easily added and has a clear predictive value [235]. Thus, aEEG could be a promising prognostic tool in clinical trials, testing potential regenerative therapies (Table 2). Overview of clinical and preclinical studies of diagnostic modalities (biomarker, imaging) for vulnerable neonates born preterm.

### 5.3. Postnatal Factors

Following birth, preterm infants receive supportive care, including adequate respiratory support, blood pressure support, normalization of fluid-electrolyte balance and blood glucose levels, feeding, and, if necessary, antibiotic treatment and treatment of seizures [254].

#### 5.3.1. Ventilation-Induced Brain Injury

Ventilation is a key aspect of the postnatal management of preterm neonates. Clinical advances such as antenatal steroids and surfactant replacement therapy have proven successful in preventing respiratory distress syndrome and improved postnatal survival. In addition, antenatal steroids are associated with a reduced risk for intraventricular hemorrhage [255]. However, repeated antenatal dosing is still under debate as it is associated with decreased fetal growth and does not reduce mortality. In addition, data on long-term adverse effects are currently lacking [256]. Nonetheless, a large proportion (92%) of preterm neonates, especially those younger than 28 weeks gestational age, cannot maintain proper oxygen levels and require respiratory support, albeit mostly non-invasively (e.g., continuous positive airway pressure) [227]. Still, much low birth weights for gestational age neonates will initially require mechanical ventilation.

Mechanical ventilation in preterm infants is associated with increased risk for brain injury and subsequent adverse neurodevelopmental outcomes, especially when pre-existing inflammation is present [257]. Mechanisms considered responsible are induction of a systemic immune response and cardiopulmonary and hemodynamic instability leading to a neuroinflammatory response [257,258]. Collectively, mechanical ventilation, even at low tidal volumes or for short periods, is an important factor in the etiology of preterm brain injury and should be taken into account when designing (pre)clinical studies.

#### 5.3.2. Feeding

During the last trimester of pregnancy, the fetal undergoes rapid growth; total brain volume increases by 140% over these 10 weeks [259]. During this period of rapid growth, the fetal brain is particularly vulnerable to nutrient deficiencies. Antenatal exposure to undernutrition has been associated with long-lasting adverse neurodevelopmental outcomes [260]. Greater postnatal intake of specific nutrients, as reviewed by Hortensius and colleagues, is positively related to improved neurodevelopmental outcomes, including fractional anisotropy, higher motor, and cognitive scores and improved white matter integrity [259].

#### 5.3.3. Postnatal Medication

Although often lifesaving, many of the medications administered to preterm infants, such as corticosteroids and narcotics have known adverse effects on the CNS and may compromise neurodevelopment. Postnatal administration, especially of potent synthetic glucocorticoids is associated with impaired long-term neurodevelopment [261]. Importantly, the administration of hydrocortisone as an alternative for dexamethasone did not appear to be associated with neurological deficits [262]. As such, the effects of prenatal (as briefly discussed in Section 5.3.1) and postnatal administration of specific medications should be taken into account when modeling preterm brain injury, especially for long-term studies.

## 6. Conclusions

The etiology of preterm brain injury is complex and multifactorial. In this review, we mainly focused on antenatal risk factors that directly cause preterm brain injury and potentially sensitize to injury when combined with additional triggers around the time of birth. Preterm infants are the most vulnerable of babies and additional adverse events after birth are compounded by clinical interventions such as mechanical ventilation, hyperoxia as well as and neonatal infection, which contributes to disturbed brain development, white and grey matter injury, and increased predisposition to short term and long term adverse neurodevelopmental outcomes. As such, every preterm neonate has its own “fingerprint” of risk factors that potentially contribute to the development of preterm brain injury. We propose that achieving the correct combination of interventions at the correct time in relation to the onset, nature, and stage of injury will be of great clinical importance to support optimal neurodevelopmental outcomes. We propose that future studies should (1) address the multifactorial nature of preterm brain injury, including postnatal hits, thereby allowing (2) appropriate patient stratification based on reliable biomarkers that provide insight into disease onset and progression, and can also be used to monitor therapeutic treatment efficacy. (3) In this regard, a clinical and experimental focus on understanding the impact of dose, route, and timing of treatment is paramount; (4) to explore synergistic protection treatments, as a single therapy is unlikely to provide complete protection or endogenous repair required for normal neurodevelopmental outcome; and (5) to complete long-term follow-up on these studies.

## Figures and Tables

**Figure 1 cells-09-01871-f001:**
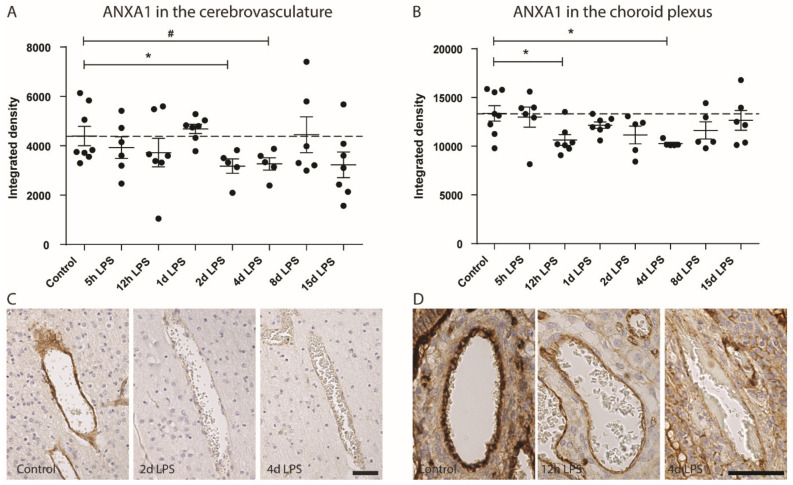
Annexin A1 (ANXA1) expression over time in response to intra-amniotic lipopolysaccharide (LPS) exposure in the vasculature of the fetal brain parenchyma and the choroid plexus. (**A**) Integrated density (mean grey value of stained area*percentage of the stained area) of ANXA1 immunoreactivity in the cerebrovasculature. (**B**) The integrated density of ANXA1 expression within the vasculature of the choroid plexus. (**C**) Representative picture of ANXA1 expression within the blood vessels within the brain parenchyma of coronal sections containing the hippocampus; 200× magnification, scale bar 50 µm. (**D**) Representative picture of ANXA1 within the blood vessels of the choroid plexus; 400× magnification, scale bar 50 µm. Data are presented as mean/standard error of the mean and the Kruskal–Wallis test was performed: *p* < 0.05 = *, *p* < 0.1 = #.

**Table 1 cells-09-01871-t001:** Overview of the clinical and preclinical studies for treatment (cell-based, antibodies, drugs) of vulnerable preterm neonates.

	Setup	Model/Clinical Setting	Main Findings	Reference
**Hypothermia**	**Meta-analysis** 11 randomized controlled trials (>35 weeks GA)	HI + HT	- HT ↓mortality or major neurodevelopmental disability to 18 months of age and in survivors - HT should be instituted in term and late preterm infants with moderate-to-severe hypoxic-ischemic encephalopathy if identified before 6 h of age	[129]
	**Clinical** 208 infants >35 weeks GA	HI + HT	- HT ↓ death or cases of an IQ score below 70 (47% vs. 58%;) - HT ↓ death (28% vs. 44%) - HT ↓ death or severe disability (41% vs. 60%)	[130]
	**Clinical** 314 infants >35 weeks GA	HI + HT	- The positive predictive value of a severely abnormal aEEG assessed by the voltage and pattern methods was 0.63 and 0.59 respectively in non-cooled infants and 0.55 and 0.51 in cooled infants	[131]
	**Clinical** 31 infants 34–35 weeks GA	HI + HT	- Hypothermia-associated complications in 90% of infants - More prevalent white matter injury in cooled preterm infants compared to cooled term neonates	[132]
	**Preclinical** Preterm (0.7 GA) ovine fetuses	HI + HT	- HT protects O4+ cells - HT ↓ of microglia and Ki67+ cells in the periventricular white matter	[133]
	**Preclinical** Preterm (0.7 GA) ovine fetuses	HI + HT	- HT ↑ EEG spectral frequency, but not total intensity - HT ↓ loss of immature oligodendrocytes in periventricular white matter - HT ↓ neuronal loss in hippocampus and basal ganglia - HT ↓ of microglia and caspase-3	[74]
	**Experimental** P7 rat	LPS i.p. + left carotid artery occlusion + HT	- HT not protective in LPS-sensitized brains	[134]
	**Experimental** P7 rat	PAM3CSK4 i.p. + left carotid artery occlusion + HT	- HT neuroprotective in presence of Gram-positive infection	[135]
**Stem Cells**	**Experimental** P5 rat	Intracranial ibotenate + hUCB-MNCs i.p./i.v.	- hUCB-MN i.p. did not reach systemic circulation; high amounts induced a detrimental systemic and cerebral inflammatory response - hUCB-MN i.v. no effect on lesion size, microglial activation, astroglial cell density, cell proliferation in white matter or cortical plate	[136]
	**Preclinical** Preterm (0.7 GA) ovine fetuses	HI + MSC i.v.	- MSC i.v. ↓ microglial proliferation, ↓ loss of oligodendrocytes, ↓ demyelination, ↓ electrographic seizure activity, peripheral T-cell tolerance, ↓T-cell invasion into the brain	[65]
	**Preclinical** Preterm (0.7 GA) ovine fetuses	HI + MAPC i.v.	- MAPC i.v. ↓number of seizures, prevented decrease in baroreceptor reflex sensitivity after global HI, ↓ microglial proliferation, prevention of hypomyelination, modulation of the peripheral splenic inflammatory response	[67]
	**Preclinical** Preterm (0.7 GA) ovine fetuses	HI + hAECs i.n.	- hAECs↑ brain weight, ↑restoration of immature/mature OLs and ↑ myelin basic protein, ↓ microglia and astrogliosis, partially improved cortical EEG frequency distribution, ↓ loss of cortical area,↓ cleaved-caspase-3 expression, ↑neuronal survival in deep grey matter nuclei	[73]
	**Experimental** P4 Rat	IVH +UCBC i.c.v.	- UCBCs ↓ post-hemorrhagic hydrocephalus development, ↓astrogliosis, ↓cell death, ↓expression of pro-inflammatory cytokines in CSF, ↑corpus-callosal thickness, ↑myelin basic protein expression, ↑behavioral tests vs. IVH group	[137]
	**Clinical trial** (NCT02274428, phase I), IVH; mean GA 26.1 ± 0.7 weeks	IVH + UCBC i.c.v.	- No infant died or showed serious adverse effects related with stem cell transplantation	[138]
	**Experimental** P7 rat	Unilateral HI + MAPC i.v./i.c.v.	- i.v. or intracerebral MAPC ↑ motor and neurologic score, hippocampal cell preservation vs. veh group	[139]
**Extracellular Vesicles**	**Experimental**, P3 rat	LPS i.p. + MSC-EV i.p.	- MSC-EV ↓ inflammation-induced neuronal and cellular degeneration, astrogliosis and microgliosis, ↑ restoration of short-term and long-term microstructural abnormalities, long-lasting cognitive functions	[140]
	**Preclinical** Preterm (0.7 GA) ovine fetuses	HI + MSC-EV i.v.	- MSC-EV ↑ brain function measured by ↓ total numbers and duration of seizures, preserved baroreceptor reflex sensitivity, tendency to prevent hypomyelination	[66]
	**Experimental** P2 rat	LPS i.p. + HI + MSC i.n.	- MSC-EVs reach frontal part of the brain after 30 min and distribute within 3 h over the whole brain - MSC-EV↓ neuronal cell death, ↑myelination, oligodendrocyte and neuron cell counts, neurodevelopmental outcome vs. perinatal brain injury	[141]
**Cytokine Treatment**	**Preclinical** Preterm (0.86 GA) ovine fetuses	Carotid occlusion + anti-IL-6 antibody i.v.	- Anti-IL-6 antibody ↓ cerebral IL-6 protein concentrations, ↑ BBB integrity and modulates tight-junction proteins	[142]
	**Preclinical** Preterm (0.7 GA) ovine fetuses	LPS i.v. + TNF-α antibody i.v.	- Anti-TNF-α delayed the rise in circulating IL-6, prolonged the increase in IL-10, attenuated EEG suppression, hypotension and tachycardia after LPS boluses, ↓ gliosis, ↑ TNF-positive cells, proliferation, total oligodendrocytes	[143]
	**Preclinical** Preterm (0.86 GA) ovine fetuses	Carotid occlusion + anti-IL-1β antibody i.v.	- Anti-IL-1β antibody ↓ cerebral IL-1β protein concentrations (P < 0.03) and protein expressions (*p* < 0.001), ↑ BBB integrity (*p* < 0.04)	[144]
	**Preclinical** Preterm (0.86 GA) ovine fetuses	Carotid occlusion + anti-IL-1β antibody i.v.	- Anti-IL-1β antibodies ↓ (*p* < 0.05) the global pathological injury scores: number of apoptotic positive cells/mm^2^, caspase-3 activity - Anti-IL-1β antibodies does not affect myelination and astrogliosis	[145]
	**Preclinical** Preterm (0.86 GA) ovine fetuses	Carotid occlusion + anti-IL-1β antibody i.v.	- Anti-IL-1β antibody ↓ blood-to-brain transport of radiolabeled IL-1β (*p* < 0.04) across brain regions	[146]
**hrEPO**	**Experimental** P7 rat	Unilateral HI + hrEPO i.p.	- hrEPO ↑revascularization in the ischemic hemisphere, ↓infarct volume, ↑neurological outcome, neurogenesis in the subventricular zone and migration of neuronal progenitors into ischemic cortex and striatum	[147]
	**Experimental** P7 rat	Unilateral HI + hrEPO i.p.	- hrEPO ↑ oligodendrogenesis and maturation of oligodendrocytes, neurogenesis, behavioral neurological outcome 14 d after HI, ↓ white matter injury - hrEPO did not reduce brain volume loss	[148]
	**Experimental** P7 rat	Unilateral HI + hrEPO s.c.	- hrEpo treatment ↓ brain injury, apoptosis, and gliosis, in a dose-dependent U-shaped manner at both 48 h and one week - Three doses of hrEPO (5000 U/kg) was optimal providing maximal benefit with limited total exposure	[149]
	**Experimental** P7 rat	Unilateral HI + hrEPO i.p.	- Therapeutic benefit of EPO when given immediately following induction of HI injury - Diminished benefit from a 60-min-delayed injection of EPO and no protection following a 180-min-delayed injection	[150]
	**Experimental** P5 mouse	PVL (ibotenic acid intra-cranial) + EPO	- Single dose of EPO is sufficient to reduce excitotoxic brain injury Therapeutic window of <4 h for EPO - Minor hematopoietic effects were observed following EPO treatment	[151]
	**Preclinical** Preterm (0.7 GA) ovine fetuses	HI (UCO) + hrEPO i.v.	- hrEPO ↓reduced neuronal loss, numbers of caspase-3-positive cells in the striatal caudate nucleus, CA3 and dentate gyrus of the hippocampus, and thalamic medial nucleus↑ total, but not immature/mature oligodendrocytes in white matter tracts, cell proliferation, ↓ microgliosis and astrogliosis, seizure burden with more rapid recovery of electroencephalogram power, spectral edge frequency, and carotid blood flow	[62]
	**Preclinical** Preterm (0.7 GA) ovine fetuses	LPS i.v. + hrEPO i.v.	- hrEPO did not improve fetal hypoxemia, hypotension induced by LPS - hrEPO ↓ brain injury ↑ myelination in the corticospinal tract and the optic nerve	[152]
	**Clinical trial** NCT02036073; 800 infants ≤32-weeks	Extreme/very preterm + hrEPO	- rhEPO ↓ rate of moderate/severe neurological disability in the placebo group (22 of 309, 7.1% vs. 57 of 304, 18.8%, *p* < 0.001) - Repeated low-dose rhEPO treatment ↓ the risk of long-term neurological disability in very preterm infants with no obvious adverse effects	[153]
	**Clinical trial** NCT0137827; extremely preterm infants, (*n* = 741)	Extreme preterm + hrEPO	- No significant difference between the EPO group and the placebo group in the incidence of death or severe neurodevelopmental impairment at two years of age (97 children (6%) vs. 94 children (26%); relative risk, 1.03; 95% CI, 0.81 to 1.32; *p* = 0.80) - High-dose EPO treatment administered to extremely preterm infants from 24 h after birth through 32 weeks of postmenstrual age did not result in a lower risk of severe neurodevelopmental impairment or death at two years of age	[154]

Abbreviations: aEEG: amplitude-integrated electroencephalography, CI: confidence interval, d: day(s), EEG: electroencephalography, GA: gestational age, HI: hypoxia-ischemia, HT: hypothermia, hAECs: human amnion epithelial cells, h: hour(s), hrEPO: human recombinant erythropoietin, UCB-MNCs: human umbilical cord blood mononuclear cells, IL: interleukin, i.c.v.: intracerebroventricular, i.n.: intra nasal, i.p.: intraperitoneal, i.v.: intravenous, NT: normothermia, TNF: tumor necrosis factor, veh: vehicle.

**Table 2 cells-09-01871-t002:** Overview of the clinical and preclinical studies of diagnostic modalities (biomarker, imaging) for vulnerable neonates born preterm.

	Setup	Model/Clinical Setting	Main Findings	Reference
**Plasma Biomarkers**	**Clinical** *n* = 155 <37 weeks GA Cord blood IL-6	Preterm birth	- Cord blood IL-6 concentrations > 11 pg/mL associated with increased rate of severe neonatal morbidity	[43]
	**Clinical** 94 patients with preterm labor Amniotic fluid IL-6, TNF-α, IL-1β	Preterm birth	- IL-6, TNF-α, IL-1β ↑ in amniotic fluid of preterm neonates with white matter injury vs. preterm neonates without white matter lesions - ↑ of IL-6, TNF-α, IL-1β levels still significant when adjusted for GA and birth weight	[44]
	**Clinical** 315 patients with preterm labor (20–35 weeks’ GA) Cord blood IL-6	Preterm birth	- A cord blood IL-6 ≥17.5 pg/mL has a sensitivity of 70% and a specificity of 78% in the identification of funisitis	[229]
	**Preclinical** Intra-amniotic LPS exposure of fetal sheep IL-6, IL-8 (0.7–0.8 GA)	Inflammation (LPS exposure)	- Fetal plasma IL-6 transiently increased from 5 h until 24 h after intra-amniotic LPS exposure. - Fetal plasma IL-8 increased at four- and eight-days post-intra-amniotic LPS	[56]
**VOC**	**Preclinical** Ovine chorioamnionitis model (0.8 GA)	Infection (Ureaplasma parvum)	- Changes in the VOC profile of ewes are detectable with good accuracy > 72 h post-infection - VOC profile changed as the duration of infection progresses	[230]
**Imaging**	**Clinical** Infants < 31 weeks GA (*n* = 77) MRI	Very preterm	- Identification of infants with abnormal motor outcome based on the FA data from early MRI with mean sensitivity 70%, mean specificity 74%, mean AUC 72%, mean F1 score of 68% and mean accuracy 73%. - Identification patches around the motor cortex and somatosensory regions with high precision (74%). - Part of the cerebellum, and occipital and frontal lobes were also highly associated with abnormal NSMDA/motor outcome.	[231]
**EEG**	**Clinical** Very low birth weight infants (*n* = 95)	Extreme preterm/very preterm	- No significant difference between conventional EEG amplitude and intensity for infants with or without evidence of white matter injury - Premature infants with increasingly severe white matter injury had progressively lower SEFs vs. infants who did not exhibit white matter injury → SEF-based measures are useful for defining the presence and severity of white matter injury	[232]
	**Clinical** Median GA 25 weeks (22–30 weeks; *n* = 94)	Extreme preterm/very preterm	- Poor outcome was associated with depressed aEEG/EEG already during the first 12 h and with prolonged interburst intervals and higher interburst percentage at 24 h - Long-term outcome can be predicted by aEEG/EEG with 75–80% accuracy already at 24 postnatal hours in very preterm infants, also in infants with no early indication of brain injury	[233]
	**Clinical** GA between 27 and 32 weeks (*n* = 12)	Very preterm/moderate preterm	- Absent cyclicity on aEEG within 24 h of age is associated with poor outcome in preterm infants	[234]
	**Clinical** GA < 30 weeks GA (*n* = 65)	Very preterm	- Data-driven approach to quantify EEG maturational deviations in preterms with normal and abnormal neurodevelopmental outcomes. - Abnormal outcome trajectories were associated with clinically defined dysmature and disorganized EEG patterns.	[235]

Abbreviations: aEEG: amplitude-integrated electroencephalography, AUC: area under the curve, CUS: cerebral ultrasound, EEG: electroencephalography, GA: gestational age, h: hour(s), IL: interleukin, MRI: Magnetic resonance imaging, NSMDA: neurosensory motor developmental assessment, PPROM: preterm premature rupture of the membranes, SEF: spectral edge frequency, TNF: tumor necrosis factor, VOC: volatile organic compounds.

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
