# Peer review of "Preterm Brain Injury, Antenatal Triggers, and Therapeutics: Timing Is Key"

_cells, 2020, doi:10.3390/cells9081871_

Round 1

Reviewer 1 Report

The Review manuscript titled “Preterm Brain Injury, antenatal triggers and therapeutics: Timing is key” by Ophelders et al provides a review of the latest literature on two important antenatal risk factors for preterm brain injury – chorioamnionitis and hypoxia-ischemia. The review is well written and provides a description of latest literature on this subject matter. Some issues in this review are:

  • In the abstract, introduction and throughout the manuscript the authors should be more specific about preterm birth. Many of the significant neurodevelopmental delays and brain imaging findings are associated with extremely preterm birth and extremely low birth weight.
  • Abstract lines 20-21 the sentence structure needs to be modified (In the majority substantial….).
  • Line 40: Cortical myelination begins at approximately 34 weeks of gestations? Reference (s)? May need to include a sentence stating that cortical brain regions myelinate at different time points.
  • Line 58: capital P – “ The Precise mechanisms…”
  • In section 1.2: Clarification is necessary. Does the “neuronal phenotype” refer to GABAergic interneurons? The authors state that they are resistant to cell death. If this is referring to GABAergic interneurons, are they referring to the interneurons in the subcortical white matter? In the cortical regions such as prefrontal cortex it has been shown that there are decreased interneurons and decreased GABA on MR spectroscopy.
  • In Section 2.3, can the authors clarify what are the three stages of brain injury and its timing? The description of each of the phases is appropriate and thorough.
  • Clarification is section 2.3: It is the author’s opinion that the therapeutic window (critical window) of opportunity for neuroprotection is in the first phase?
  • Regarding section 2.3: Do the authors think that the therapeutic window for neuro-rehabilitation must commence in the first phase or is it possible that rehabilitation can be initiated at any time point?
  • Section 3.2: While experimental evidence in preclinical models of preterm brain injury from HI demonstrates that therapeutic hypothermia is protective, this paragraph might need a sentence to clarify that therapeutic hypothermia is not standard of care for infants born very preterm.
  • Line 229: Chen et al reference? Number?
  • Are the panels in Figure 1 published elsewhere or is this new original data? Reference in figure legend may be necessary if not original.
  • Section 4, lines 372-381: It should also be stated that we are limited in identifying all the infants born preterm that have brain injury. There is a need for biomarkers or specific studies that will identify these children.
  •  Section 5.2.2 Imaging: The authors should include when MR imaging of the brain is performed? Is there a precedent to image earlier – during the preterm period?
  • Figure 2: The overall concept of the figure is appropriate and good. However, in this reviewers opinion the figure can be confusing. The risk factors, biomarkers timing and therapy are all noted with a structure in the background that appears to be shaped as a brain. The confusion is that the way the figure is presented it appears that it is denoting specific brain regions versus concepts.  

Reviewer 2 Report

This is a thorough and well-written review of the antenatal etiopathogenic mechanisms of brain damage in preterm infants. In its current version, however, the manuscript seems a bit like an item list, and could be further improved by addressing the following points: 
  • please add deeper clinical comments and considerations about the list of issues addressed, e.g., the feasibility of the proposed biomarkers, practical limitations of some of the mentioned therapeutic strategies, etc. 
  • when evidence on available or proposed treatments is mentioned, it should be always specified if it is pre-clinical or clinical and, in the latter case, if it based on term or preterm neonates.
  • although post-natal mechanisms of injury do not lie into the paper purposes, I think they would deserve to be briefly described in a dedicated paragraph.
  • I strongly suggest summarizing the mentioned evidence in specific tables, in order to improve the data presentation and its readability. For example, different tables related to biomarkers or treatments may be built, specifying such information as the study population (clinical/preclinical, term/preterm, sample size), reference values (if available), treatment safety (if available).  
  • In its actual form, the paper figure looks like a mix of several different data joint together.  I would create an additional figure illustrating cell-specific mechanisms of injury and which type of parenchyma do they involve. 
  • Section 2, title: perhaps "pathophysiology of preterm brain damage" better fits the contents of this section. 
